# 72-Hour transport recovery of antimicrobial resistant *Neisseria gonorrhoeae* isolates using the InTray® GC method

**Keely S. Paris**[1]*, **Brandon Font**[2], **Sanjay R. Mehta**[3], **Irvin Huerta**[2], **Claire C. Bristow**[3]*

**1** University of California San Diego, La Jolla, CA, United States of America, **2** BioMed Diagnostics Inc., Research & Development, White City, Oregon, United States of America, **3** Division of Infectious Diseases and Global Public Health, Department of Medicine, University of California San Diego, La Jolla, CA, United States of America

* keelyparis@gmail.com (KSP); cbristow@health.ucsd.edu (CCB)

## Abstract

Recovery of *Neisseria gonorrhoeae* isolates exposed to a range of transport times and temperatures was quantitatively assessed for two transport devices, BioMed Diagnostics' InTray GC® and Copan Diagnostics' Liquid Amies Elution Swab (ESwab®) Collection and Transport System. Respective devices were inoculated with *N. gonorrhoeae*, exposed to simulated transport conditions and spread-plated from serial dilutions in duplicates onto chocolate agar in order to count CFU (colony-forming units) in the range of 25–250. Baseline CFU/mL averages of time-zero transport for each device was compared to either 24 hour (Eswab) or 72 hour (InTray GC) CFU/mL average to assess recovery of six clinical isolates of *N. gonorrhoeae*, and differences showing no greater than a 3 $\log_{10}$ (± 10%) decline between comparative time points qualified as acceptable. Our findings suggest that the InTray GC system has the potential to transport clinical isolates for ≤72 hours with acceptable *N. gonorrhoeae* recovery.

## Introduction

*Neisseria gonorrhoeae* is a gram-negative obligate human pathogen and the etiological agent of gonorrhea. Gonorrhea is not typically fatal, but if left untreated, can lead to various symptoms and significant complications, such as pelvic inflammatory disease, sterility, ectopic pregnancy, or, rarely, sepsis or death [1]. After chlamydia, gonorrhea is the second most commonly reported notifiable infection in the U.S. and has experienced a 63% increase in cases since 2014, including a 5.0% increase in incidence in 2018 alone [2]. As of 2016, the global incidence of infection with *N. gonorrhoeae* was 87 million cases per year [3].

Some *N. gonorrhoeae* strains have quickly developed resistance to many available antibiotics, often limiting the available treatment options. A single 500 mg dose of intramuscular ceftriaxone is the CDC-recommended first-line drug therapy for uncomplicated anogenital and pharyngeal gonorrhea in adults [4]. In the United States, antibiotic resistance trends are monitored by the CDC's Gonococcal Isolate Surveillance Program (GISP), which has used resistance surveillance data since the 1980s to stop recommending penicillin, tetracycline, fluoroquinolones, cefixime, and other oral cephalosporins to treat gonorrhea [5]. In recent

**Data Availability Statement:** All relevant data are within the manuscript and its Supporting Information files.

**Funding:** This study was supported by National Institute of Allergy and Infectious Diseases in the form of a grant awarded to CCB and KSP (K01AI136725) and BioMed Diagnostics Inc. in the form of funding for all study procedures and salaries for BF and IH. The specific roles of these authors are articulated in the 'author contributions' section. BioMed Diagnostics Inc. had a role in the decision to publish, the data collection, and in the study design in order to ensure FDA standards were met, but the funders did not have any additional role in the data analysis or preparation of the manuscript.

**Competing interests:** The authors have read the journal's policy and have the following competing interests: BF and IH are employees of Biomed Diagnostics Inc., which provided funding for all study procedures. InTray® GC is a registered trademark of BioMed Diagnostics Inc., which also owns a patent for the InTray® GC platform. This does not alter our adherence to PLOS ONE policies on sharing data and materials.

years, strains with high resistance to ceftriaxone have arisen in several countries, severely limiting treatment options for patients infected with these particularly resistant strains [6].

*N. gonorrhoeae* testing and detection often involves nucleic acid amplification testing (NAAT) which, although highly sensitive and specific, does not allow for antibiotic susceptibility testing. The benefits of NAAT, particularly increased sensitivity when screening urine, urogenital and pharyngeal swabs, have led to increased popularity and the subsequent decrease in bacterial culturing in clinical settings [7]. However, culturing is necessary not only to test susceptibility to antibiotics, but also to monitor treatment failures and identify outbreaks of antimicrobial resistant *N. gonorrhoeae* strains. *N. gonorrhoeae* cultures require specific conditions to optimize their survival, including being inoculated onto nutritive and selective chocolate agar and incubated from 35–37˚ in a $CO_2$ enriched environment [5]. The difficulty of culturing *N. gonorrhoeae* makes it difficult to perform antibiotic susceptibility panels, which can further decrease how often they are used.

With ceftriaxone resistance and gonorrhea incidence both on the rise, antibiotic resistance monitoring is more important than ever. Public health efforts must focus on developing new, more advanced technologies to monitor resistance of *N. gonorrhoeae*, particularly to ceftriaxone and other higher generation cephalosporins. Due to the difficulty of growing and recovering *N. gonorrhoeae*, growth media and transport systems need to be evaluated for efficacy and must be optimized for maximum recovery and quality control. Prior studies have examined the difference in viability between different commercially available transport systems and generally found that the InTray GC®, a $CO_2$ generating packaging medium manufactured by BioMed Diagnostics, had a high rate of recovery after extended transport times [7]. Additionally, InTray GC plates have a shelf life of approximately one year, so are well-suited for use in settings where standard chocolate agar plates may not be widely available or viable after storage [8]. In an effort to build upon these previous studies, we assessed bacterial recovery of clinical isolates of *N. gonorrhoeae* in transport conditions in the InTray GC 72 hours after an initial period of incubation. The purpose of this study was also to receive approval from the U.S. Food and Drug Administration (FDA) to modify the intended use to extend the allowable times that the InTray GC device could be used to transport specimens; the methods were chosen to demonstrate viability based on FDA standards.

## Materials and methods

We used *N. gonorrhoeae* AR Bank isolates #0165, #0181, #0197, #0202, and #0175 from the CDC and FDA Antibiotic Resistance Isolate bank [9]. We also used isolate ATCC 43069 from the nonprofit company American Type Culture Collection, located in Manassas, VA, USA. The minimum inhibitory concentrations to several antibiotics, as well as the results to a β-lactamase test, are recorded for each isolate in S1 Table. Each strain was visually confirmed to be *N. gonorrhoeae* following initial recovery from the suppliers. These six strains were chosen to represent a diverse set of MIC profiles.

The experimental groups consisted of *N. gonorrhoeae* inoculated onto the InTray GC transport devices (BioMed Diagnostics, Inc., OR, USA), and onto the Liquid Amies Elution Swab (ESwab) Collection and Transport System (Copan Diagnostics, CA, USA). Inoculum verification was also performed on standard chocolate agar plates to determine the viability of the experimental strains.

Colony forming units (CFUs) were calculated using spectrophotometers at 625nm adjusted to a 0.5 McFarland standard in 0.85% physiological saline, as per CSLI guidelines [10]. We calculated the mean $\log_{10}$ CFU/mL for each time point by taking the average of the duplicates of each dilution (dilutions were plated twice prior to calculating CFUs for that specific dilution), then averaging all the dilutions within each triplicate (three plates [A, B, and C] for each time).

From there, we averaged the $\log_{10}$ CFU/mL of all three triplicates to create a single value for the time point. Percent decrease was calculated for each strain in both experimental groups by subtracting the final $\log_{10}$ CFU/mL value from the initial $\log_{10}$ CFU/mL value, dividing this value by the initial $\log_{10}$ CFU/mL quantity, then multiplying by 100%. Strain failure was determined by a 100% decrease in viable colonies after 72 hours in the InTray GC group and 24 hours in the ESwab group. If the strain failed, the procedure was repeated. Different time periods were chosen for each experimental group based on the intended use for each device; the ESwab is approved for use in transporting *N. gonorrhoeae* for up to 24 hours, while the InTray GC is currently approved for up to 48 hours, and in this study we assess the InTray GC for viability in transporting specimen for up to 72 hours [11,12]. Acceptance criteria for quantitative recovery, as well as experimental procedures for inoculation, strain viability verification, length of transport time and plating were based on CLSI M40-A2 Quality Control of Microbiological Transport Systems; Approved Standard 2nd edition [10].

First, we created source tubes for each respective isolate (AR 0165, ATCC 43069, AR 0181, AR 0197, AR 0202, AR 0175) by separately inoculating standard chocolate agar plates with each isolate, then incubating each at 37˚C in a 5–7% $CO_2$ environment for 24 hours (Fig 1). We then picked colonies from individual plates with a $1\mu L$ loop and mixed each into separate vials of 1.2mL of a 0.85% saline solution, before measuring the mean average absorption with a blanked spectrophotometer at 625nm adjusted to a 0.5 McFarland standard (S2 Table). This process was completed twice for each isolate to create two source tubes, which we then used to make 10-fold and 2-fold serial dilutions from each to achieve concentrations of $10^7$, $10^6$, $10^5$, $10^4$, $10^3$, $5 \times 10^2$, $2.5 \times 10^2$, and $10^2$ CFU/mL.

For each individual isolate, we then inoculated 9 InTray GC plates (triplicates from 3 manufacturing lots; A, B, and C) with $20\mu L$ from the $10^3$ dilution according to manufacturer instructions, before placing them into a 37˚C incubator (Fig 1). 24 hours later, we removed the plates, counted and recorded the colonies on each, then chose the six most homogenous plates (2 from each lot A, B, and C). The two most similar counts from each lot were picked for downstream analysis and the higher count was tagged as plate time-zero InTray GC to avoid over-estimation viability bias for InTray GC time-72 hours. We plated the 3 InTray GC time-zero plates by scraping and removing all colonies from each plate (starting with A) with a $1\mu L$ loop and swirling into 1mL of 0.85% saline, then made serial dilutions in the range of $10^4$ to $10^0$ CFU/mL (S2 Table). After, we plated $100\mu L$ from each dilution in duplicates onto standard chocolate agar plates and placed them into the $CO_2$ enriched incubator. At the same time, we placed the three InTray GC time-72 hours plates into a therapak-biohazard-specimen transport bag under controlled room temperatures. Temperature was monitored and documented over the 72-hour time period for all strains. For strain AR 0197 and AR 0202, we placed the specimen transport bag in the trunk of a car and monitored temperature changes with a USB data logger to simulate real-world transport conditions. AR 0202 failed in the uncontrolled temperature environment of the car, thus it was repeated at controlled room temperature. We then removed these plates after 72 hours and repeated the plating and dilution process that we used for time-zero InTray GC, then placed these plates into the $CO_2$ enriched incubator. After incubating both the time-zero InTray GC and the time-72 hour InTray GC, we removed them from the incubators and recorded the plate counts in the range of 25–250 CFUs.

In addition to the InTray GC, we also tested Copan ESwabs' viability over time. For each isolate, we began by inoculating 3 sets of triplicate ESwabs (time-zero A, B, C; time-24 hours A, B, C) by placing each swab into a separate test tube with $100\mu L$ of the $10^7$ CFU/mL dilution for 10–15 seconds, before replacing each swab into their respective container of Liquid Amies (S2 Table). The triplicates of ESwab time-24 were then placed into specimen transport bags, under similar conditions to InTray GC time-72 hour. We gave the triplicates of ESwab time-zero 5–15 minutes

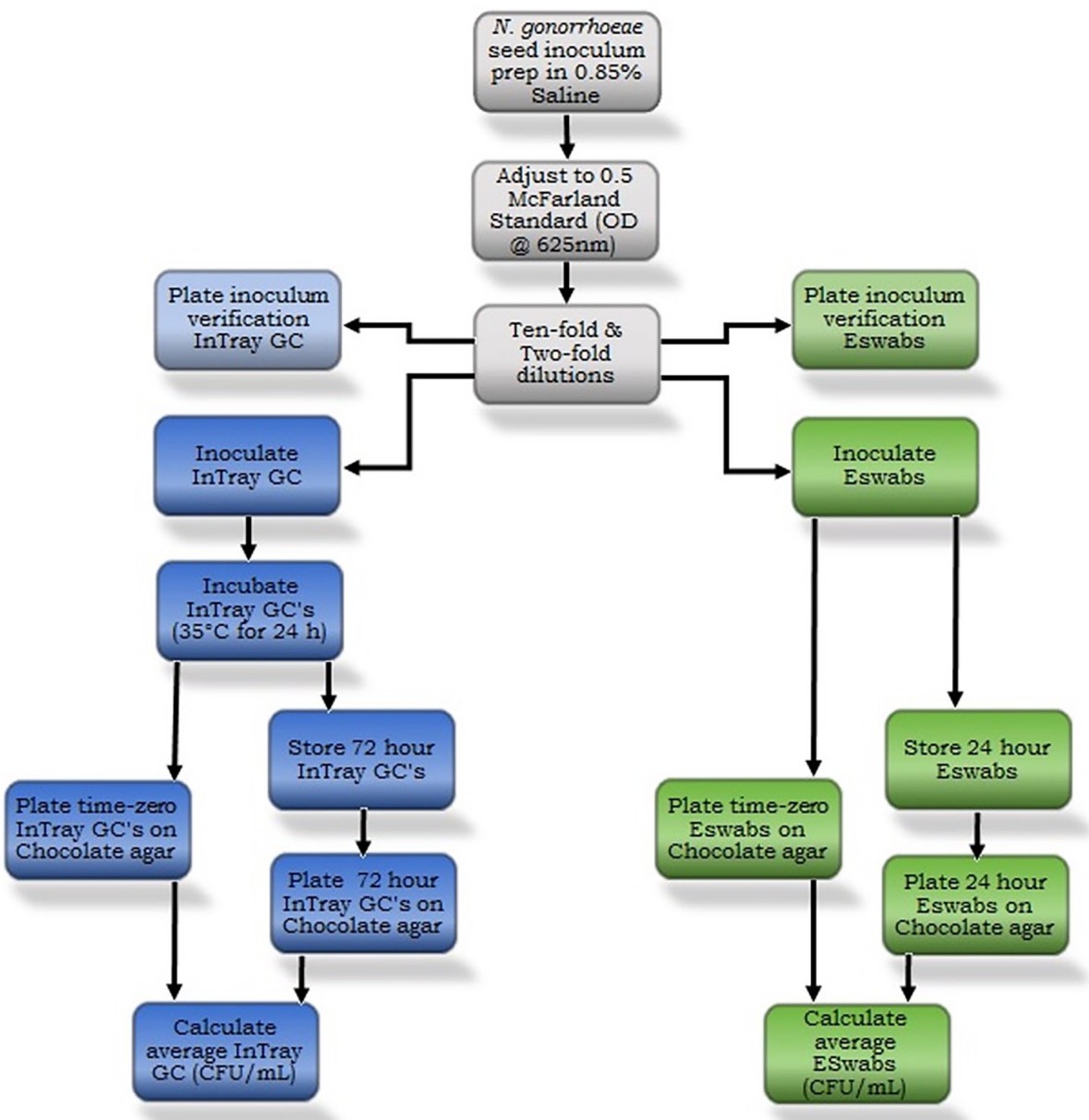

**Fig 1. Methods flowchart for the InTray GC and ESwab transport recovery of *Neisseria gonorrhoeae* isolates.**

to rest before we vortexed each and expressed the $100\mu L$ of liquid from each swab into separate 5mL test tubes. Then, we made serial dilutions ($10^4$, $10^3$ and $10^2$ CFU/mL) for each set (A, B, and C), plated $100\mu L$ onto standard chocolate agar plates in duplicates, and placed these into the $CO_2$ incubator. 24 hours later, the plates were removed from the incubator and the CFU per plate was recorded. The same day, we removed the time-24 hours plates from the specimen transport bag and repeated the plating, dilution, and incubation process from ESwab time-zero.

In order to demonstrate strain viability, we also performed inoculum verification for the InTray GC and ESwab. For the InTray GC, we spread $100\mu L$ from the $10^4$, $10^3$, $5 \times 10^2$, $2.5 \times 10^2$, and $10^2$ CFU/mL dilutions onto duplicate standard chocolate agar plates (10 plates total) and incubated them in the $CO_2$ enriched incubator for 24 hours. After 24 hours, we removed

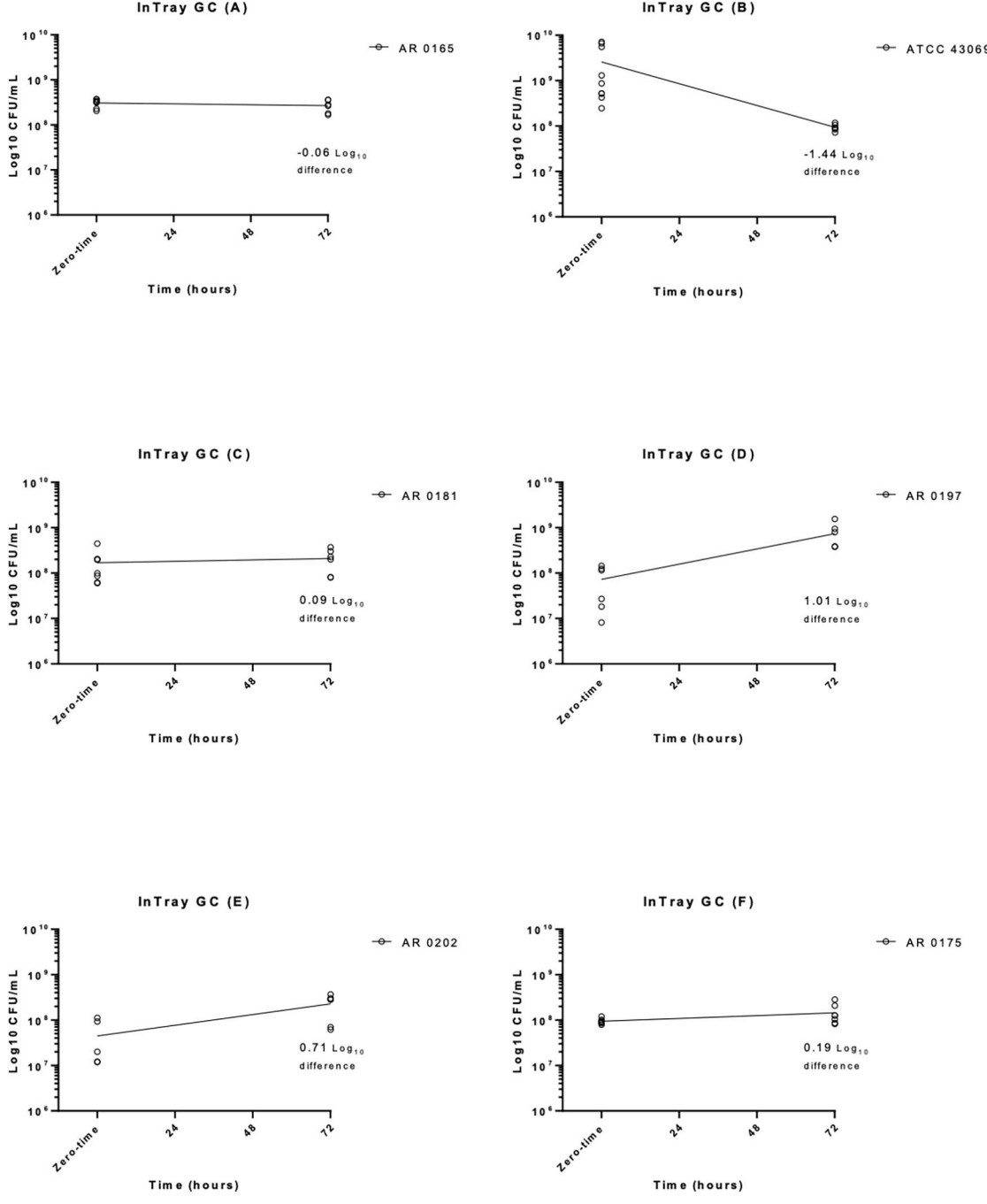

**Fig 2. Time in hours versus Log₁₀ CFU/mL for the InTray GC for all six strains.** CFU; colony forming units.

the inoculum verification plates and recorded each duplicate's plate count in the range of 25–250 CFU per plate.

## Results

At 72 hours, out of the 6 strains tested on the InTray GC, 5 strains passed and 1 strain initially failed before being repeated at controlled room temperatures (AR 0202) (Fig 2). AR 0165

**Table 1. Quantitative recovery results of the subject InTray GC transport device for six *Neisseria gonorrhoeae* type culture strains.**

| *Neisseria gonorrhoeae* type culture strain | InTray GC replicate | CFU/mL Recovery: time-zero control group | | CFU/mL Recovery: 72-hour transport group | | Results | |
|---|---|---|---|---|---|---|---|
| | | *N | Mean ± SD CFU/mL | *N | Mean ± SD CFU/mL | Log$_{10}$ Difference | Interpretation |
| CDC/FDA AR#O165 | A<br>B<br>C | 6<br>4<br>4 | $3.1 \times 10^8 \pm 6.7 \times 10^7$ | 4<br>2<br>3 | $2.7 \times 10^8 \pm 8.4 \times 10^7$ | -0.06 | Acceptable Recovery |
| ATCC® 43069 | A<br>B<br>C | 2<br>8<br>8 | $2.6 \times 10^9 \pm 2.9 \times 10^9$ | 4<br>0<br>6 | $9.3 \times 10^7 \pm 1.6 \times 10^7$ | -1.44 | Acceptable Recovery |
| CDC/FDA AR#O181 | A<br>B<br>C | 8<br>4<br>4 | $1.7 \times 10^8 \pm 1.3 \times 10^8$ | 4<br>4<br>4 | $2.1 \times 10^8 \pm 1.2 \times 10^8$ | 0.09 | Acceptable Recovery |
| CDC/FDA AR#O197 | A<br>B<br>C | 4<br>6<br>2 | $7.3 \times 10^7 \pm 6.1 \times 10^7$ | 6<br>2<br>4 | $7.4 \times 10^8 \pm 4.7 \times 10^8$ | 1.01 | Acceptable Recovery |
| CDC/FDA AR#O202 | A<br>B<br>C | 2<br>2<br>4 | $4.5 \times 10^7 \pm 4.5 \times 10^7$ | 4<br>4<br>4 | $2.3 \times 10^8 \pm 1.3 \times 10^8$ | 0.71 | Acceptable Recovery |
| CDC/FDA AR#O175 | A<br>B<br>C | 4<br>6<br>4 | $9.4 \times 10^7 \pm 1.3 \times 10^7$ | 4<br>6<br>4 | $1.5 \times 10^8 \pm 7.4 \times 10^7$ | 0.19 | Acceptable Recovery |

Differences in mean average CFU/mL concentrations were analyzed by subtracting the Log$_{10}$ CFU/mL average of the 72-hour groups from the Log$_{10}$ CFU/mL average of the time-zero control groups. Each group consisted of three InTray GC devices labeled A, B & C, as listed. N is equal to the number of recovery quantitation plates in the accepted countable 25–250 CFU range.

experienced a 12.9% decrease in log$_{10}$ CFU/mL (0.06 log$_{10}$ reduction) from 24 to 72 hours, and ATCC 43069 had a larger percent decrease of 96.37% (1.44 log$_{10}$ reduction). The other four strains actually had increases in their log$_{10}$ CFU/mL over the same time period, with AR 0181 having a 18.72% increase (0.09 log$_{10}$ increase); AR 0197 increased by 90.23% (1.01 log$_{10}$ increase); AR 0202 experienced a slightly smaller percent increase of 80.5% (0.71 log$_{10}$ increase); isolate AR 0175 increased by 35.43% (0.19 log$_{10}$ increase) over the 72 hours. When averaged, there was a 19.27 percent increase (0.08 log$_{10}$ increase) across successful strains after 72 hours on the InTray GC, which demonstrated acceptable recovery (Table 1).

For the 6 strains tested on the ESwab, after 24 hours, 5 strains passed and the same strain that failed for the InTray GC also initially failed for the ESwab before being repeated at controlled room temperatures (AR 0202) (Fig 3). After 24 hours on the ESwab, AR 0165 decreased in log$_{10}$ CFU/ml by 94.5% (1.26 log$_{10}$ reduction), ATCC 43069 decreased by 98.30% (1.77 log$_{10}$ reduction), AR 0181 decreased by 99.22% (2.11 log$_{10}$ reduction), AR 0197 decreased by 87.70% (0.91 log$_{10}$ reduction), AR 0202 decreased by 99.69% (2.51 log$_{10}$ reduction), and AR 0175 decreased by 98.42% (1.80 log$_{10}$ reduction). When averaged, the percent decrease across successful strains after 24 hours on the ESwab, was 96.31% (1.73 log$_{10}$ reduction) (Table 2).

Strain AR 0197, which was stored in the trunk of a car with a USB data logger, experienced a maximum temperature of 33.5° Celsius, a minimum temperature of 2.5° Celsius, and an average of 15.8°C (Std: 8.3°C) (S1 Fig). Strains AR 0165, AR 0181, and ATCC 43069 were stored under ambient conditions, with a maximum temperature of 22.5° Celsius, a minimum

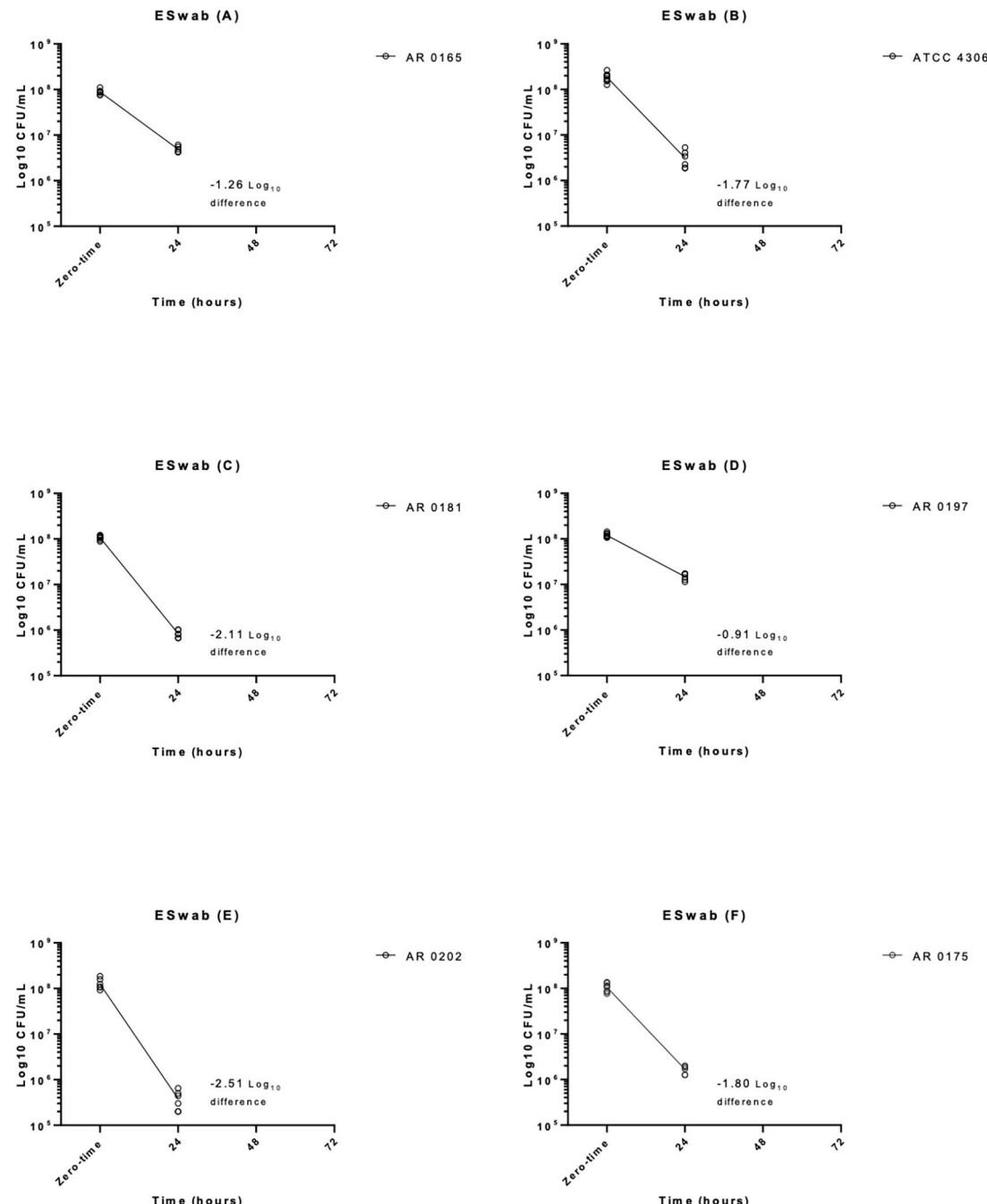

**Fig 3. Time in hours versus Log10 CFU/mL for the Copan ESwab for all six strains.** CFU; colony forming units.

of 19˚ Celsius, and an average of 20.2˚ Celsius (Std: 0.6˚ Celsius) (S2 Fig). Strains AR 0202 and AR 0175 were also stored under ambient conditions, with an average temperature of 23.1˚ Celsius (Std 0.2˚ C), a maximum temperature of 24˚ Celsius, and a minimum temperature of 23˚ Celsius (S3 Fig).

All inoculum verification tests confirmed strain viability of the experimental groups, including for strain AR 0202, which initially failed for all experimental groups before being repeated.

**Table 2. Quantitative recovery results of the ESwab transport device for six *Neisseria gonorrhoeae* type culture strains.**

| *Neisseria gonorrhoeae* type culture strain | ESwab replicate | CFU/mL Recovery: time-zero control group | | CFU/mL Recovery: 24-hour transport group | | Results | |
|---|---|---|---|---|---|---|---|
| | | *N | Mean ± SD CFU/mL | *N | Mean ± SD CFU/mL | Log$_{10}$ Difference | Interpretation |
| CDC/FDA AR#O165 | A<br>B<br>C | 6<br>4<br>4 | $8.8. \times 10^7 \pm 1.2 \times 10^7$ | 2<br>2<br>2 | $4.9 \times 10^6 \pm 7.8 \times 10^5$ | -1.26 | Acceptable Recovery |
| ATCC® 43069 | A<br>B<br>C | 8<br>6<br>6 | $1.8 \times 10^8 \pm 3.9 \times 10^7$ | 2<br>2<br>4 | $3.1 \times 10^6 \pm 1.4 \times 10^6$ | -1.77 | Acceptable Recovery |
| CDC/FDA AR#O181 | A<br>B<br>C | 4<br>6<br>6 | $1.1 \times 10^8 \pm 1.1 \times 10^7$ | 2<br>2<br>2 | $8.4 \times 10^5 \pm 1.6 \times 10^5$ | -2.11 | Acceptable Recovery |
| CDC/FDA AR#O197 | A<br>B<br>C | 6<br>6<br>6 | $1.2 \times 10^8 \pm 1.2 \times 10^7$ | 2<br>2<br>2 | $1.5 \times 10^7 \pm 2.5 \times 10^6$ | -0.91 | Acceptable Recovery |
| CDC/FDA AR#O202 | A<br>B<br>C | 4<br>4<br>6 | $1.2 \times 10^8 \pm 3.5 \times 10^7$ | 2<br>2<br>2 | $3.8 \times 10^5 \pm 1.8 \times 10^5$ | -2.51 | Acceptable Recovery |
| CDC/FDA AR#O175 | A<br>B<br>C | 6<br>6<br>4 | $1.1 \times 10^8 \pm 2.2 \times 10^7$ | 2<br>2<br>2 | $1.7 \times 10^6 \pm 3.4 \times 10^5$ | -1.80 | Acceptable Recovery |

Differences in mean average CFU/mL concentrations were analyzed by subtracting the Log$_{10}$ CFU/mL average of the 24-hour groups from the Log$_{10}$ CFU/mL average of the time-zero control groups. Each group consisted of three ESwab devices labeled A, B & C, as listed. N is equal to the number of recovery quantitation plates in the accepted countable 25–250 CFU range.

## Discussion

After 72 hours in the InTray GC transport systems, there was a small percent increase in viable colonies of *N. gonorrhoeae* (Table 1), while the loss of colonies after only 24 hours of transport using the Copan ESwab was fairly high. The minimal decrease in bacterial quantities showed the InTray GC's potential to maintain a transport environment that facilitates the survival of various clinical isolates of *N. gonorrhoeae* over increased transport times. After 72 hours, the InTray GC plates experienced an average 19.27% increase in colonies, while the ESwab lost 24.82% of colonies after 24 hours in the same conditions. Strain AR 0197, which experienced more drastic temperature changes due to being stored in the trunk of a car, had an increase of 90.23% in log$_{10}$ CFU/ml for the InTray GC plates, and a decrease of 87.70% for the ESwab plates. We hypothesize that isolate AR 0202 initially failed all experiments due to overheating, as we initially stored the 72-hour experiment in the trunk of a car to simulate transport in uncontrolled temperatures.

These data support a previous study examining four commercial transport systems and comparing recovery rates between the systems, which found that the InTray GC had optimal recovery of specimen in comparison to other systems [7]. Our study builds upon this research by demonstrating that the InTray GC plates could be subjected to extended transport times up to 72 hours without significant loss of specimen viability. This project was subject to limitations, including the limited number of isolates tested. However, the purpose of this study was

to demonstrate acceptable recovery on the InTray GC device for an FDA intended use submission for which the 6 isolates are sufficient. The strain growth in terms of percent recovery for four of the strains on the InTray GC (AR 0181, AR 0197, AR0202, AR0175) could also potentially be confounded by the initial bacterial load in each sample. However, based on prior studies examining recovery on the InTray GC, we did not expect to see vast declines between time points, and rather expected to see small $\log_{10}$ increases and decreases [7].

This research is integral to future research and monitoring of antibiotic resistance in *N. gonorrhoeae*. The potential of the InTray GC to recover bacteria after long periods of transport should ideally lead to its increased use as a transport system for clinical isolates. With an average of 2.84% of colonies lost after 72 hours, the InTray GC could be used to transport longer distances to facilities with advanced antibiotic susceptibility testing capabilities. In order to help manage *N. gonorrhoeae* infections and associated antibiotic resistance testing, microbiology laboratories should utilize transport systems that maintain viable clinical isolates during extended transport times. Considering the difficulty in maintaining viable *N. gonorrhoeae* cultures in transport conditions, as well as the rising rates of antibiotic resistance, it is important to recognize the InTray GC as a potentially viable option for a transport system with high recovery rates after extended periods of transport time.

## Supporting information

**S1 Fig. Time in hours stored in the trunk of a car versus temperature in degrees Celsius for strain AR 0197.** Mean temperature: 15.8˚C (Std: 8.3˚C), Maximum temperature: 33.5˚C, Min temperature: 2.5˚C.
(TIFF)

**S2 Fig. Time in hours versus temperature in degrees Celsius for strains AR 0165, AR 0181, and ATCC 43069.** Mean temperature: 20.2˚C (Std: 0.6˚C), Maximum temperature: 22.5˚C, Min temperature: 19˚C.
(TIFF)

**S3 Fig. Time in hours versus temperature in degrees Celsius for strains AR 0202 and AR 0175.** Mean temperature: 23.1˚C (Std: 0.2˚C), Maximum temperature: 24˚C, Min temperature: 23˚C.
(TIFF)

**S1 Table. Minimum inhibitory concentrations (MIC) (in μg/ml) of experimental isolates to azithromycin, cefixime, cefpodoxime, ceftriaxone, ciprofloxacin, penicillin, and tetracycline, as well as the results to a β -lactamase test.** MIC; minimum inhibitory concentration. ATCC 43069 does not have antibiotic susceptibility MIC results.
(DOCX)

**S2 Table. Inoculation density and procedure adjustments applied to CLSI M40-A2 section 8.11 Swab Elution Method (Quantitative) for the Eswab and InTray transport GC devices.**
(DOCX)

## Author Contributions

**Formal analysis:** Keely S. Paris, Sanjay R. Mehta, Claire C. Bristow.

**Investigation:** Brandon Font, Irvin Huerta.

**Methodology:** Brandon Font, Irvin Huerta.

**Writing – original draft:** Keely S. Paris.

**Writing – review & editing:** Keely S. Paris, Brandon Font, Sanjay R. Mehta, Claire C. Bristow.

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
