## [Decision Letter · Decision Letter 0]

12 Jul 2021

PONE-D-21-16017

72-Hour Transport Recovery of Antimicrobial Resistant Neisseria gonorrhoeae Isolates Using the InTray® GC Method

PLOS ONE

Dear Dr. Keely S Paris,

Your manuscript "72-Hour Transport Recovery of Antimicrobial Resistant Neisseria gonorrhoeae Isolates Using the InTray® GC Method" have been assessed by external reviewers and me. Although the data of potential interest, the paper is unfortunately not acceptable for publication in the present form. There are several essential issues as outlined by reviewers that will need to be addressed. The reviewers are overall positive that with significant revision, the quality of the paper may be improved. Therefore, I invite you to revise your paper, considering the points raised during the review process.

Please go over your manuscript text and ensure that it has been written concisely and clearly. At the same time, we ask you to make sure your manuscript complies with our format requirements detailed on the journal website. 

We look forward to receiving your revised manuscript.

Kind regards,

**Dr. Supram Hosuru Subramanya, Ph.D.**

Academic Editor

PLOS ONE

Journal Requirements:

"CCB and KSP acknowledge funding from National Institute of Allergy and Infectious Diseases (K01AI136725). All study procedures were funded and conducted by BioMed Diagnostics Inc. (https://biomeddiagnostics.com/). "

We note that one or more of the authors have an affiliation to the commercial funders of this research study : BioMed Diagnostics Inc.

2.1. Please provide an amended Funding Statement declaring this commercial affiliation, as well as a statement regarding the Role of Funders in your study. If the funding organization did not play a role in the study design, data collection and analysis, decision to publish, or preparation of the manuscript and only provided financial support in the form of authors' salaries and/or research materials, please review your statements relating to the author contributions, and ensure you have specifically and accurately indicated the role(s) that these authors had in your study. You can update author roles in the Author Contributions section of the online submission form.

2.2. Please also provide an updated Competing Interests Statement declaring this commercial affiliation along with any other relevant declarations relating to employment, consultancy, patents, products in development, or marketed products, etc.  

Reviewers' comments:

Reviewer's Responses to Questions

**Comments to the Author**

1. Is the manuscript technically sound, and do the data support the conclusions?

Reviewer #1: Yes

Reviewer #2: Yes

Reviewer #3: Yes

2. Has the statistical analysis been performed appropriately and rigorously? 

Reviewer #1: Yes

Reviewer #2: Yes

Reviewer #3: Yes

3. Have the authors made all data underlying the findings in their manuscript fully available?

Reviewer #1: Yes

Reviewer #2: Yes

Reviewer #3: Yes

4. Is the manuscript presented in an intelligible fashion and written in standard English?

Reviewer #1: Yes

Reviewer #2: Yes

Reviewer #3: Yes

5. Review Comments to the Author

Reviewer #1: Gonorrhea incidence is on the rise as is the level of antibiotic resistance in the circulating strains. It is important to develop techniques that maintain viability during transport of this organism. In this study, Paris et. al assessed bacterial recovery of clinical isolates of N. gonorrhoeae in transport conditions using InTray GC (the current standard developed by BioMed Diagnostics), 72 hours after an initial period of incubation. The purpose of this study was also to receive approval from the US FDA to modify the intended use to extend the allowable times that the InTray GC device could be used to transport specimens. The study is rigorous and well written, however there are some concerns that must be addressed:

1. The central finding in this paper is that the recovery is good after 72h in InTrayGC. However, the fact that most of the strains seem to “grow” (in terms of % recovered) during the 72 hours in the InTrayGC (the strain that grew the most, 0197, increased by 1 log) could likely confound any quantitation of the initial bacterial load that was present in the sample. While it is evident that the authors are focused on recovery, this confounder should be at least addressed in the discussion.

2. Continuing from above, the different strains grow to different extents. What do the authors make of this finding?

3. L78- It is not clear why the authors chose to test the particular strains #0165, #0181, #0197, #0202, and #0175. While they provide different MIC values for antibiotics in Table S1 to demonstrate the differences in the strains, how do the authors deflect the criticism that the strains were “cherry picked”? Some reasoning for why they chose these strains would be useful information to provide. Additionally, highlighting the unbiased nature of the process that lead to the selection of these 5-6 strains would allay some of the skepticism associated with strain selection (perhaps they are phylogenetically well spaced out on a tree? Or this is part of a standard set of strains used to test any new method in the field?).

4. L 97- To an uninitiated reader, it is not clear why the 72-hour time point was chosen for InTray GC and 24 hours for Eswab. It would help if appropriate literature describing the current standard in the field was cited.

5. L180 – “the same strain failed”. The word “as” is a typo

6. Figure 3 does not need the 48 and 72 hr timepoints on the X axis

Reviewer #2: The manuscript by Paris and colleagues investigates the recovery of Neisseria gonorrhoeae from two transport systems. The BioMed Diagnostics InTray GC system and Copan Diagnostics Liquid Amies Elution Swab (ESwab) were inoculated with a standard inoculum of six different N. gonorrhoeae strains (five clinical isolates from the CDC FDA specimen bank and one ATCC strain) and held at various times and temperatures to simulate transport conditions from a clinic to a laboratory. There was a 19.3% increase in CFU/ml from GC InTray plates and a 24.8% decrease in CFU/ml from the Eswab system at 72 hours. The data confirms previous studies reporting similar findings and will be used by BioMed Diagnostics to seek a modification from the US Food and Drug Administration to extend the allowable times that the GC InTray can be used to transport clinical specimens.

Comments

General comment: Review the manuscript for consistent use of Neisseria gonorrhoeae and N. gonorrhoeae. The full bacterial name should only be spelled out when it is first used in the manuscript and subsequently referred to as N. gonorrhoeae. There is no need to put N in paratheses as written on line 20.

Line 53: Does the sentence refer to culture being necessary to identify outbreaks of antimicrobial resistant (AMR) N. gonorrhoeae or simply gonorrhea outbreaks? Non-culture methods would be more useful for gonorrhea outbreaks and culture being need for AMR outbreaks.

General comment for the Methods section: Were the strains confirmed to be N. gonorrhoeae following initial recovery from the supplier? The methods used to confirm the identity of the strains should be added.

Reviewer #3: REVIEW NOTES FOR MANUSCRIPT NUMBER- PONE-D-21-16017

The manuscript is well and clearly written, with figures and tables that are easy to comprehend. It describes a research study, assessing the recovery of six Neisseria gonorrhoeae clinical isolates that are inoculated into two different transport devices, the BioMed Diagnostics’ InTray GC and the Copan Diagnostics’ Liquid Amies Elution Swab (ESwab) Collection and Transport System.

The approach used is appropriate. Assessment of recovery from both systems was done after 24h for the ESwab and after 72h for the BioMed Diagnostics’ InTray GC. Each system was assigned to a separate experimental group while a third group consisted of a control group made up of the same clinical samples inoculated unto standard chocolate agar plates to confirm viability of the clinical isolates. Both systems were subjected to the same temperature conditions. The ability of each system to maintain viability of the isolates was determined by comparing the CFU/ml at initial time zero and the CFU/ml at the end point time.

The ability of the InTrayGC system to maintain and transport viable gonococcal isolates for recovery after more than 24hrs has been confirmed by earlier studies (Papp et al., 2016; Beverly et al., 2000). However, the work under review was done to obtain approval from the FDA to use the InTrayGC system to transport clinical specimen for an extended period (up to 72hrs) before recovery of isolates of Neisseria gonorrhoeae. The methods used were chosen to demonstrate viability based on FDA standards.

The study results show that the InTrayGC system can maintain N. gonorrhoeae clinical isolates for up to 72h for recovery of viable isolates.

This study is important in that Neisseria gonorrhoeae is a very fastidious organism that is easily lost to temperature fluctuations during transportation of clinical specimens. There is a need to develop and assess appropriate transport systems to overcome this weakness.

ABSTRACT AND INTRODUCTION

The main research question and the key findings of this study are clearly summarized. Other current literature on the topic has been referenced

FIGURES AND TABLES

All the associated text supports the information in the tables and figures. The figures are clear and readable, captions are complete and accurate.

METHODS

The experiments used are appropriate and the data is enough to support the conclusions that were arrived at, although the assessment of more specimen would have given more weight to the data. However, the author may need to explain why the InTray GC system was incubated here. The system is described as a transport system, when in actuality, incubating the system before transportation has a strong positive influence on isolate survival/viability.

Limitations were addressed adequately; data was collected and interpreted correctly. This study can be replicated.

RESULTS, DISCUSSION AND CONCLUSIONS

The results, discussion and conclusion are well described and clearly written

MINOR COMMENTS/EDITS

Line 66. Space between the end of the sentence and the parenthesis ie. (7) is missing

Line 193. Space between the & and C is missing

Line 261. (ref 6.). The reference title was not complete

Line 233. Typo at “used”, it should be “use”

6. PLOS authors have the option to publish the peer review history of their article (what does this mean?). If published, this will include your full peer review and any attached files.

Reviewer #1: **Yes: **aditya bandekar

Reviewer #2: No

Reviewer #3: No

---

## [Author Response · Author response to Decision Letter 0]

22 Sep 2021

Thank you for reviewing our paper for publication in PLOS ONE. Enclosed is a revision version of our manuscript, ‘72-Hour Transport Recovery of Antimicrobial Resistant Neisseria gonorrhoeae Isolates Using the InTray® GC Method'. We found reviewer comments insightful and constructive and have revised the manuscript according to their feedback. Below is a detailed account of the revisions made in response to reviewer comments:

Reviewer #1: Gonorrhea incidence is on the rise as is the level of antibiotic resistance in the circulating strains. It is important to develop techniques that maintain viability during transport of this organism. In this study, Paris et. al assessed bacterial recovery of clinical isolates of N. gonorrhoeae in transport conditions using InTray GC (the current standard developed by BioMed Diagnostics), 72 hours after an initial period of incubation. The purpose of this study was also to receive approval from the US FDA to modify the intended use to extend the allowable times that the InTray GC device could be used to transport specimens. The study is rigorous and well written, however there are some concerns that must be addressed:

Thank you for your time in reviewing our manuscript. We appreciate the feedback provided. 

1. The central finding in this paper is that the recovery is good after 72h in InTrayGC. However, the fact that most of the strains seem to “grow” (in terms of % recovered) during the 72 hours in the InTrayGC (the strain that grew the most, 0197, increased by 1 log) could likely confound any quantitation of the initial bacterial load that was present in the sample. While it is evident that the authors are focused on recovery, this confounder should be at least addressed in the discussion.

This is a great question. Based on previous experiments examining recovery on the InTray GC (Papp, 2016, Fig 1 A&B), we did not expect to see vast declines between time points, and rather expected to see small log10 differences up or down. For this reason, we were not particularly concerned over bacterial growth during the 72-hour time period. We also performed some unpaired T-tests for the averages between the time-zero and 72 hour groups, though we opted not to put this data into the main manuscript as it was somewhat external to what our study was examining. These tests, however, did seem to support our expectations for small increases and decreases in log10 quantities. To enhance the clarity of our manuscript, we added the following sentences to the discussion starting at line 293: “The strain growth in terms of percent recovery for four of the strains on the InTray GC (AR 0181, AR 0197, AR0202, AR0175) could also potentially be confounded by the initial bacterial load in each sample. However, based on prior studies examining recovery on the InTray GC, we did not expect to see vast declines between time points, and rather expected to see small log10 increases and decreases”. 

2. Continuing from above, the different strains grow to different extents. What do the authors make of this finding?

Thank you for your question. Strain AR0197 was exposed to increased temperature (as described in the supplemental file) and we hypothesize there was, in reality, cell growth on the nutritive InTray GC (this is reflected in the small P-value of 0.0059 for this strain from the unpaired t-tests mentioned above). While other strains showed slight log10 increases or decreases, the P-values are non-significant (ranging from 0.10 to 0.55). We suspect the largest decline from the ATCC 43069 strain was due to a “laboratory” adaptation effect. This is an old strain in the lab and we hypothesize it has adapted to growing quite well on media at warm temperatures, but when exposed to cooler temperatures it did not survive as well as the other strains. For strain AR 0202, we would hypothesize that this strain had the ability to grow at lower temperatures based on the small P value of 0.0083.

3. L78- It is not clear why the authors chose to test the particular strains #0165, #0181, #0197, #0202, and #0175. While they provide different MIC values for antibiotics in Table S1 to demonstrate the differences in the strains, how do the authors deflect the criticism that the strains were “cherry picked”? Some reasoning for why they chose these strains would be useful information to provide. Additionally, highlighting the unbiased nature of the process that lead to the selection of these 5-6 strains would allay some of the skepticism associated with strain selection (perhaps they are phylogenetically well spaced out on a tree? Or this is part of a standard set of strains used to test any new method in the field?).

Thank you for this question. Five of the 6 strains we used in our study were from the CDC’s AR isolate bank. We chose these strains for several reasons, the primary one being that these strains had antibiotic resistance patterns that were most likely to be encountered in the field. By choosing strains based on the current epidemiology of N. gonorrhoeae, we hoped to demonstrate the InTray GC’s viability in transporting common strains with a variety of MIC profiles. Furthermore, we used strains with different MIC profiles to ensure a diverse sample. Also, by retrieving all but one strain from the CDC’s AR isolate bank, we tried to keep the sample as standardized as possible. At line 108, we added the sentence “These six strains were chosen to represent a diverse set of MIC profiles”. 

4. L 97- To an uninitiated reader, it is not clear why the 72-hour time point was chosen for InTray GC and 24 hours for Eswab. It would help if appropriate literature describing the current standard in the field was cited.

Thank you for your comment. The reason 24 hours was chosen for the ESwab is because it’s current intended use states that it can be used to transport specimen for up to 24 hours, while the InTray can be used for up to 48 hours (although the purpose of this manuscript is to extend the allowable transportation time to 72 hours in the intended use). We added the following to line 129: “Different time periods were chosen for each experimental group based on the intended use for each device; the ESwab is approved for use in transporting specimen for up to 24 hours, while the InTray GC is currently approved for up to 48 hours, and in this study we assess the InTray GC for viability in transporting specimen for up to 72 hours.” We also added the intended use for both the InTray GC and the ESwab as new references. 

5. L180 – “the same strain failed”. The word “as” is a typo

Thank you for pointing this out. We have made the appropriate changes.

6. Figure 3 does not need the 48 and 72 hr timepoints on the X axis

Thank you for your comment. To keep the figures consistent between the ESwab and the InTray GC, we have opted to keep the 48 and 72 hour timepoints in figure 3. While the ESwab and InTray GC are testing different timepoints for recovery, we believe it would aid in clarity to keep the figures consistent so readers can easily compare recovery rates between the two. 

Reviewer #2: The manuscript by Paris and colleagues investigates the recovery of Neisseria gonorrhoeae from two transport systems. The BioMed Diagnostics InTray GC system and Copan Diagnostics Liquid Amies Elution Swab (ESwab) were inoculated with a standard inoculum of six different N. gonorrhoeae strains (five clinical isolates from the CDC FDA specimen bank and one ATCC strain) and held at various times and temperatures to simulate transport conditions from a clinic to a laboratory. There was a 19.3% increase in CFU/ml from GC InTray plates and a 24.8% decrease in CFU/ml from the Eswab system at 72 hours. The data confirms previous studies reporting similar findings and will be used by BioMed Diagnostics to seek a modification from the US Food and Drug Administration to extend the allowable times that the GC InTray can be used to transport clinical specimens.

Thank you for reviewing our manuscript. We appreciate your feedback.

Comments

General comment: Review the manuscript for consistent use of Neisseria gonorrhoeae and N. gonorrhoeae. The full bacterial name should only be spelled out when it is first used in the manuscript and subsequently referred to as N. gonorrhoeae. There is no need to put N in paratheses as written on line 20.

Thank you for your comment. We have updated the manuscript accordingly.

Line 53: Does the sentence refer to culture being necessary to identify outbreaks of antimicrobial resistant (AMR) N. gonorrhoeae or simply gonorrhea outbreaks? Non-culture methods would be more useful for gonorrhea outbreaks and culture being need for AMR outbreaks.

We were referencing outbreaks of AMR N. gonorrhoeae. To improve the clarity of the manuscript, we edited the sentence beginning on line 71 to say “However, culturing is necessary not only to test susceptibility to antibiotics, but also to monitor treatment failures and identify outbreaks of antimicrobial resistant N. gonorrhoeae strains”. 

General comment for the Methods section: Were the strains confirmed to be N. gonorrhoeae following initial recovery from the supplier? The methods used to confirm the identity of the strains should be added.

Each strain was visually confirmed to be N. gonorrhoeae (gray/white, small, translucent, convex, shiny). In the materials and methods section, starting at line 108, we added the following sentence: “Each strain was visually confirmed to be N. gonorrhoeae following initial recovery from the suppliers.”

Reviewer #3: REVIEW NOTES FOR MANUSCRIPT NUMBER- PONE-D-21-16017

The manuscript is well and clearly written, with figures and tables that are easy to comprehend. It describes a research study, assessing the recovery of six Neisseria gonorrhoeae clinical isolates that are inoculated into two different transport devices, the BioMed Diagnostics’ InTray GC and the Copan Diagnostics’ Liquid Amies Elution Swab (ESwab) Collection and Transport System.

The approach used is appropriate. Assessment of recovery from both systems was done after 24h for the ESwab and after 72h for the BioMed Diagnostics’ InTray GC. Each system was assigned to a separate experimental group while a third group consisted of a control group made up of the same clinical samples inoculated unto standard chocolate agar plates to confirm viability of the clinical isolates. Both systems were subjected to the same temperature conditions. The ability of each system to maintain viability of the isolates was determined by comparing the CFU/ml at initial time zero and the CFU/ml at the end point time.

The ability of the InTrayGC system to maintain and transport viable gonococcal isolates for recovery after more than 24hrs has been confirmed by earlier studies (Papp et al., 2016; Beverly et al., 2000). However, the work under review was done to obtain approval from the FDA to use the InTrayGC system to transport clinical specimen for an extended period (up to 72hrs) before recovery of isolates of Neisseria gonorrhoeae. The methods used were chosen to demonstrate viability based on FDA standards.

The study results show that the InTrayGC system can maintain N. gonorrhoeae clinical isolates for up to 72h for recovery of viable isolates.

This study is important in that Neisseria gonorrhoeae is a very fastidious organism that is easily lost to temperature fluctuations during transportation of clinical specimens. There is a need to develop and assess appropriate transport systems to overcome this weakness.

Thank you for your time in reading and reviewing our manuscript and for your positive and constructive comments. 

ABSTRACT AND INTRODUCTION

The main research question and the key findings of this study are clearly summarized. Other current literature on the topic has been referenced

Thank you.

FIGURES AND TABLES

All the associated text supports the information in the tables and figures. The figures are clear and readable, captions are complete and accurate.

Thank you for your feedback.

METHODS

The experiments used are appropriate and the data is enough to support the conclusions that were arrived at, although the assessment of more specimen would have given more weight to the data. However, the author may need to explain why the InTray GC system was incubated here. The system is described as a transport system, when in actuality, incubating the system before transportation has a strong positive influence on isolate survival/viability.

Limitations were addressed adequately; data was collected and interpreted correctly. This study can be replicated.

Thank you for your comments. As for why the InTray GC was incubated, this has to do with its current intended use, as the InTray GC is meant to be incubated for 24 to 48 hours prior to transportation (as described in the package insert). The intended use for the InTray GC has also been added as a reference. 

RESULTS, DISCUSSION AND CONCLUSIONS

The results, discussion and conclusion are well described and clearly written

Thank you for your comments.

MINOR COMMENTS/EDITS

Line 66. Space between the end of the sentence and the parenthesis ie. (7) is missing

Line 193. Space between the & and C is missing

Line 261. (ref 6.). The reference title was not complete

Line 233. Typo at “used”, it should be “use”

Thank you for this feedback. We have made all the suggested minor edits. 

Regards,

Keely Paris

---

## [Decision Letter · Decision Letter 1]

25 Oct 2021

72-Hour transport recovery of antimicrobial resistant *Neisseria gonorrhoeae* isolates using the InTray® GC method

PONE-D-21-16017R1

Dear Dr. Keely S Paris,

We’re pleased to inform you that your manuscript has been judged scientifically suitable for publication and will be formally accepted for publication once it meets all outstanding technical requirements.

Kind regards,

Supram Hosuru Subramanya, Ph.D.

Academic Editor

PLOS ONE

Additional Editor Comments (optional):

Reviewers' comments:

Reviewer's Responses to Questions

**Comments to the Author**

1. If the authors have adequately addressed your comments raised in a previous round of review and you feel that this manuscript is now acceptable for publication, you may indicate that here to bypass the “Comments to the Author” section, enter your conflict of interest statement in the “Confidential to Editor” section, and submit your "Accept" recommendation.

Reviewer #1: All comments have been addressed

Reviewer #2: All comments have been addressed

Reviewer #3: All comments have been addressed

2. Is the manuscript technically sound, and do the data support the conclusions?

Reviewer #1: Yes

Reviewer #2: Yes

Reviewer #3: Yes

3. Has the statistical analysis been performed appropriately and rigorously? 

Reviewer #1: Yes

Reviewer #2: N/A

Reviewer #3: Yes

4. Have the authors made all data underlying the findings in their manuscript fully available?

Reviewer #1: Yes

Reviewer #2: Yes

Reviewer #3: Yes

5. Is the manuscript presented in an intelligible fashion and written in standard English?

Reviewer #1: Yes

Reviewer #2: Yes

Reviewer #3: Yes

6. Review Comments to the Author

Reviewer #1: (No Response)

Reviewer #2: The revised manuscript properly addresses comments from the reviewers. This reviewer does not have any additional comments for consideration.

Reviewer #3: (No Response)

7. PLOS authors have the option to publish the peer review history of their article (what does this mean?). If published, this will include your full peer review and any attached files.

Reviewer #1: **Yes: **aditya bandekar

Reviewer #2: No

Reviewer #3: No

---

## [Editor Report · Acceptance letter]

11 Jan 2022

PONE-D-21-16017R1 

72-Hour transport recovery of antimicrobial resistant *Neisseria gonorrhoeae* isolates using the InTray® GC method 

Dear Dr. Paris:

I'm pleased to inform you that your manuscript has been deemed suitable for publication in PLOS ONE. Congratulations! Your manuscript is now with our production department. 

Kind regards, 

on behalf of

Dr. Supram Hosuru Subramanya 

Academic Editor

PLOS ONE